# What Is the Evidence for Dietary-Induced DHA Deficiency in Human Brains?

**DOI:** 10.3390/nu15010161

**Published:** 2022-12-29

**Authors:** Andrew J. Sinclair, Yonghua Wang, Duo Li

**Affiliations:** 1Department of Nutrition, Dietetics and Food, School of Clinical Sciences, Monash University, Notting Hill, VIC 3168, Australia; 2Faculty of Health, Deakin University, Burwood, VIC 3152, Australia; 3School of Food Science and Engineering, South China University of Technology, Guangzhou 510641, China; 4Institute of Nutrition & Health, College of Public Health, Qingdao University, Qingdao 266071, China

**Keywords:** docosahexaenoic acid (DHA), human, brain, deficiency of DHA, alpha-linolenic acid (ALA), deficiency of ALA, docosapentaenoic acid (22:5*n*-6), vegetarian, vegan

## Abstract

Docosahexaenoic acid (DHA) is a major constituent of neural and visual membranes and is required for optimal neural and visual function. DHA is derived from food or by endogenous synthesis from α-linolenic acid (ALA), an essential fatty acid. Low blood levels of DHA in some westernised populations have led to speculations that child development disorders and various neurological conditions are associated with sub-optimal neural DHA levels, a proposition which has been supported by the supplement industry. This review searched for evidence of deficiency of DHA in human populations, based on elevated levels of the biochemical marker of *n*-3 deficiency, docosapentaenoic acid (22:5*n*-6). Three scenarios/situations were identified for the insufficient supply of DHA, namely in the brain of new-born infants fed with high-linoleic acid (LA), low-ALA formulas, in cord blood of women at birth who were vegetarians and in the milk of women from North Sudan. Twenty post-mortem brain studies from the developed world from adults with various neurological disorders revealed no evidence of raised levels of 22:5*n*-6, even in the samples with reduced DHA levels compared with control subjects. Human populations most likely at risk of *n*-3 deficiency are new-born and weanling infants, children and adolescents in areas of dryland agriculture, in famines, or are refugees, however, these populations have rarely been studied. This is an important topic for future research.

## 1. Introduction

The brain is the most complex organ in the body, and knowledge of the structure, function, and biochemical and physiological processes in the brain are in a continuous state of discovery [1,2,3].

Docosahexaenoic acid (DHA) is a major cell membrane component of all neuronal cells and rod outer segments of the retina, and it is accepted as being essential for normal neural, auditory, olfactory, and visual function (as reviewed by Sinclair [4]). Many studies in the literature have asserted that various groups of people have low neural DHA levels, based on low levels of *n*-3 polyunsaturated fatty acids (PUFA) in red blood cells and/or plasma (as reviewed by von Schacky [5]). Furthermore, this proposition is supported by the supplement industry, which contributes to driving sales of *n*-3 PUFA supplements which exceeded USD 1.54 billion in 2020 [6]. The aim of this paper was to assess the evidence for widespread reductions of DHA in the brain, the possible causes of which are elaborated in Table 1.

Tissue DHA is derived from synthesis in the liver from α-linolenic acid (ALA), an *n*-3 essential fatty acid (EFA) [16], or directly from foods containing DHA [17]. Dietary deficiency of ALA leads to significant reductions in DHA in neural and retinal membranes and these reductions are consistently associated with reduced capacity to learn, to smell, to hear and to see [4]. One explanation for the altered physiological responses to *n*-3 PUFA deficiency is that DHA is required for the optimal function of integral membrane proteins (receptors, voltage-gated ion channels and enzymes), leading to ‘weakened/suboptimal’ signals from the initiating receptors [4]. Crawford et al. [18] posit that the last double bond in DHA is important for cell signalling in photoreceptor and neuronal cells, supporting the findings that from an evolutionary view DHA accumulates in these retinal and neural cells in all mammals rather than 22-carbon fatty acids with fewer double bonds, such as 22:5*n*-3. 

Curiously, when animals are maintained on ALA-deficient diets, the cell membranes of neural cells and rod outer segments showed accumulation of an alternative 22-carbon PUFA, docosapentaenoic acid (22:5*n*-6), such that the total 22-carbon proportion in these membranes is maintained at a somewhat constant level (Figure 1) [19,20]. Therefore, these tissues have the mechanisms to maintain 22-carbon PUFA homeostasis, and the alternative 22-carbon PUFA might act as a ‘buffer’ to prevent catastrophic effects of the absence of DHA in the brain [4,21,22]. Bourre et al. [19] suggested that a tissue biochemical marker of dietary *n*-3 deficiency was the ratio of C22:5*n*-6 to DHA. Moriguchi et al. [20] reported that the 22:5*n*-6 levels in serum, liver, brain and retina of *n*-3 deficient rats were approximately 3%, 5%, 10% and 25%, respectively, of total tissue fatty acids indicating that it should be possible to detect *n*-3 deficiency using blood analyses. In ALA-deficiency studies in animals which reported physiological changes in neural and retinal function, the 22:5*n*-6/DHA ranged from 1.7 to 9.3/1 in the brain and retina, equating to losses of DHA of between 63–90% [4]. It is relevant to ask why 22:5*n*-6 accumulates in neural and retinal tissues in *n*-3 PUFA deficiency. To induce a tissue deficiency of DHA, animals are fed diets in which the oil component is very low in ALA. Typically, oils such as safflower, peanut or corn have been used as the ALA content in these oils is usually between 0.1–0.3% total fatty acids. These oils are all highly enriched in LA with LA/ALA values typically more than 100/1. Owing to the competition between LA and ALA for the D6 desaturase (FADS2) and further competition between 22:4*n*-6 and 22:5*n*-3 for the same enzyme system [23], diets extremely rich in LA compared with ALA will lead to promotion of *n*-6 pathway over the *n*-3 pathway, resulting in increased 22:5*n*-6 levels and decreased DHA levels (Figure 1); these changes are especially evident in neural and retinal tissue. It is assumed that the 22:5*n*-6 is synthesised from LA in the liver and transported in plasma lipoproteins to the brain and retina [16].

There have been many intervention studies using *n*-3 PUFA supplements (mainly those containing eicosapentaenoic acid (EPA) + DPA*n*-3 + DHA) in various physiological states including pregnancy, in people suffering from depression, attention-deficit hyperactivity disorder, Alzheimer’s Disease, Mild Cognitive Impairment and other conditions associated with mental decline in ageing [5,24]. There has been a lack of consistency in results between these studies, with the majority showing nil effect. However, one recent report related to pre-term infants (<29 weeks) provided with an enteral emulsion fortified with DHA until 36-weeks postmenstrual age or discharge was associated with a moderately higher full-scale intelligence quotient score at 5 years of age compared with control-fed infants [25]. Additionally, many of these intervention studies have been underpowered. In the case of major depression, early trials reported beneficial effects especially with EPA, however the most recent systematic analysis suggested that at best there were small to modest non-clinical beneficial effects on depressive symptomatology [26]. Despite the relatively weak evidence for an effect of *n*-3 PUFA in relation to brain function, the sales of *n*-3 PUFA supplements throughout the world are more than USD1.54 billion [6]. 

The aim of this paper was to explore evidence for the existence of dietary induced reductions in brain DHA levels induced by a dietary *n*-3 PUFA deficiency, by systematically searching for published data on DHA and 22:5*n*-6 levels in brain (post-mortem) tissue, as well as in plasma/blood, cord blood, cord artery, adipose, and milk, as outlined in Figure 2.

## 2. Results

### 2.1. Postnatal Infant Brain Tissue Shows Evidence of Biochemical *n*-3 Deficiency

Two landmark studies reported significantly reduced brain DHA levels and significantly raised 22:5*n*-6 in newborn infants when they were fed formulas with high LA/ALA compared with breast milk. Farquharson and colleagues [27] reported brain DHA and 22:5*n*-6 levels in infants fed either breast milk or on two different formula diets with high LA/ALA ratios and lacking DHA. On each of the formula feeds, there were substantial increases in 22:5*n*-6 in the cerebral cortex over time (44 weeks postnatally), especially in the formula with the lowest ALA level (0.4% total fatty acids, LA/ALA 36:1). In the breast-fed infants, the mean DHA in the phosphatidylserine (PS) fraction was 23.5% compared with 14.4% in infants fed the lowest ALA formula; while the 22:5*n*-6 levels were 5.3% (breast-fed) and 10.4%, (lowest ALA formula-fed). Most of the formula-fed infants had undetectable levels of DHA in their adipose tissue [28], whereas all but one of the breast-fed infants had detectable levels of DHA, with mean value of 0.1% adipose TAG fatty acids. Similar data was reported by Makrides et al. [29]. In their study the breast-fed infant brain cortex DHA values increased with age from 7% at birth to >10% by 48 weeks postnatal age, while those fed formula (with LA/ALA from 9.4–11.3) remained at approximately 7% throughout this time. There was significantly more 22:5*n*-6 in the cortex of the formula-fed infants, and a significant negative correlation between brain DHA and 22:5*n*-6 was found across all samples. These two data sets indicated that after birth the infant brain is susceptible to the influence of high LA/ALA formula feeding and that such formulas were associated with reductions in neural DHA levels and showed biochemical signs of *n*-3 PUFA deficiency (i.e., elevated 22:5*n*-6 proportions). These data illustrate that high LA-low ALA formulas resulted in the same biochemical changes in brain fatty acid profiles as reported in ALA-deficient animals. In these two infant studies lasting for 44 weeks from birth, the 22:5*n*-6/DHA values were less than 1/1 compared with brain ratios reported in animal studies of ALA-deficiency of 1.2/1 to 4.5/1 (Figure 3). In other words, the DHA losses in the infants were not nearly as substantial as those reported in the animal studies which reported significant reductions in learning and memory.

Further data were reported from Tanzania on foetal brain tissue from still-born infants whose mothers consumed fish regularly [40]. They reported that 22:5*n*-6 was present in the foetal brain samples from foetuses of 14-, 22- and 35-weeks gestational age with the 22:5*n*-6/DHA ratios ranging from 0.24/1 to 0.39/1 (22:5*n*-6 levels were 1.39%, 2.38% and 2.88%, respectively, while the DHA values for these samples were 5.90%, 6.78% and 7.42%, respectively). 

### 2.2. Do Human Diets Resemble Those Used to Induce *n*-3 PUFA Deficiency?

Diets used to induce *n*-3 deficiency in animals typically have between 0.1% and 0.3% ALA of the total dietary lipid, no DHA, and between 50% and 70% LA (LA/ALA > 150:1). When provided to experimental animals, such diets induce significant depletions of neural DHA levels with time (months) or over several generations, associated with almost complete replacement of the losses in DHA with 22:5*n*-6. 

Human dietary fatty acids. 

Much has been written on the fatty acid content of human diets, ranging from the speculated palaeolithic diet with high proportions of AA, EPA, DPA*n*-3 and DHA [41] to vegan diets with no long chain (LC) *n*-3 PUFA and relatively high levels of LA and low to modest levels of ALA [42,43,44,45]. In the US, the adequate intake (AI) values for ALA are 1.6 g/day for ALA in men and 1.1 g/day in women [46]. Recommended intakes for LC*n*-3 PUFA are in the order of 250 mg/day [47]. 

Most human diets are rich in LA and low in ALA with the intake of LC *n*-3 PUFA dependent on whether fish and marine products are consumed. However, most human diets do not have dietary LA/ALA ratios as high as those in the diets used to induce *n*-3 deficiency in animals (Table 2). 

Vegan diets frequently have higher LA intakes than omnivores, however the ALA intakes vary perhaps dependent on their food supply (country of origin) [36]. A systematic review of 36 studies reported that PUFA intakes were highest in vegans (8.84% en) and lowest in meat-eaters (5.95% en) [48]. The ALA intakes, based on 12 studies, were higher in vegans (2.01 g/d), compared with vegetarians (1.78 g/d) and meat-eaters (1.38 g/d). Vegan diets typically contained no DHA.

Table 1 reports data from various countries as compiled by Harika et al. [47] and from several studies in the UK, Japan (women), India (school children) and four studies of vegans from different countries, as examples of differences between diet types and countries. The UK (omnivores) and the Japanese study reported the highest intakes of LC *n*-3 PUFA, while the Indian and vegan studies reported the lowest LC *n*-3 PUFA intakes between 0 and 10 mg/day. The highest LA/ALA ratios were in reported for Indian children (22.7/1) and Australian vegans (18.7/1).

**Table 2 nutrients-15-00161-t002:** Dietary intakes of PUFA, LA, ALA and LC *n*-3 fatty acids (data expressed as % energy for total PUFA, LA and ALA).

Reference	PUFA	LA	ALA	LA/ALA	LCn-3 Fatty Acids mg/Day
Harika et al. [47] (various countries) ^1^	4.7–8.9 ^1^	2.6–8.6	0.14–1.0	3.9–29	NR–419
Pinto et al. [43] (omnivores UK)	6.0	3.5	0.34	10.2	590
Miyake et al. [49] (pregnant women Japan)	6.7	5.4	0.9	6.3	460
Lakshmipriya et al. [50] (M & F India) ^2^	5.2	5.0	0.22	22.7	5
Pinto et al. [43] (vegans UK)	5.6	5.2	0.4	13.0	10
Agren et al. [44] (vegans Finland)	9.7	8.1	1.6	5.1	0
Mann et al. [42] (vegans Australia)	6.4	6.1	0.3	18.7	0
Chamorro et al. [45] (vegan males USA)	13.9	11.1	2.4	4.6	0

^1^ From 15 countries including Asia, Australia, Europe, North and Central America, Middle East; ^2^ Traditional vegetable oils (groundnut and sesame). NR—not recorded.

Hibbeln et al. [51] reported the PUFA content of major food commodities throughout the world and showed that LA was the major PUFA in almost every food category, except fish. Vegetable oils were invariably rich in LA with low levels of ALA. Seven oils in common use had LA/ALA greater than 50/1, however several oils had considerably lower ratios such as canola oil (2.2/1), olive oil (13/1) and soybean oil (7.5/1). The estimated LA intakes from 13 countries were reported to range from 0.90% E in the Philippines to 8.91% E in USA. The ALA intakes ranged from 0.06% E (Bulgaria) to 1.06% E (USA). The two countries with the lowest reported intakes of ALA were Bulgaria at 0.06% E [51] and Mexico at 0.14% E [47].

The situations most likely to induce a *n*-3 PUFA deficiency would be where people were wholly reliant on grain-based diets, with an absence of green leafy vegetables, ruminant meats and vegetable oils such as soybean oil, canola oil or olive oil and marine foods containing DHA, and who used vegetable oils for cooking with high LA and low ALA levels such as groundnut oil and sunflower oil; dietary data from India from subjects who predominantly used high LA, low ALA oils are shown in Table 1 and this type of diet had an LA:ALA of 22.7/1 [50].

What dietary information is missing?

What has not been explored in detail are the fatty acid proportions in the diets of people who lived in locations remote from the sea, and who existed on grains and/or starchy root vegetables with little access to green vegetables, fat (butter or veg oils), fish and who used *n*-3 PUFA deficient vegetable oils such as safflower, maize, coconut, sunflower, corn or groundnut oils. Examples might include places such as Sub-Saharan Africa, parts of the Indian sub-continent and some locations in central Europe.

### 2.3. Do Tissue Samples from Humans Show Evidence of *n*-3 PUFA Deficiency?

Readers are referred to a recent review by Burns-Whitmore et al. [46] which reported the ALA intakes in vegan diets and *n*-3 PUFA levels in blood, adipose and milk of vegans. The common feature of the blood and tissues was that the tissue levels of *n*-3 PUFA from vegans and vegetarians were significantly lower than for omnivores, however there was no evidence that 22:5*n*-6 levels exceeded those of DHA in the blood or tissues. 

#### 2.3.1. Plasma and Red Blood Cells in Adults

The diets which are most likely to lead to a deficiency of LC *n*-3 PUFA in tissues will be vegetarian or vegan diets (Table 2). Many studies of plasma, serum or red cell fatty acids have shown that the DHA levels were significantly reduced in vegans and vegetarians compared with omnivores [46] as shown in Table 3. Five studies were selected to illustrate the levels of *n*-3 PUFA of vegetarians and/or vegans from the UK, Finland and India in plasma and red blood cells. In each of these studies, the proportions of DHA were significantly reduced in the vegans compared with the omnivores. The ratio of 22:5*n*-6/DHA in the vegans was greater than the omnivores, with the values being 0.29, 0.15 and 0.26 (Table 2) in the vegans and 0.05, 0.03 and 0.12 (Table 3) in the omnivores, respectively. In the Agren et al. [44] study, the DHA in the vegans RBC-PS fraction was decreased by approximately 50% (from 14.1% to 7.8%), however the loss of DHA was not replaced by 22:5*n*-6 in this fraction (increased from 0.6% to 1.2%) (Table 3).

The plasma PL fatty acid profile of school children from India (median age 11 years) are also reported in Table 3. The children were predominantly (>96%) no*n*-vegetarians with fresh-water fish and shellfish being consumed by >80% of the children. The reported dietary intakes of ALA and DHA were 18 and 53 mg/day, respectively, for boys, and 14 and 39 mg/day, respectively, for girls [53]. The LA intakes were not reported. The plasma data showed low DHA levels (1% PL fatty acids) and a higher 22:5*n*-6/DHA of 0.6/1 compared with adult vegans shown in Table 3. 

The gaps in the literature were several. Firstly, 22:5*n*-6 levels were not always reported which could be partly due to technical issues with the chromatography, and/or lack of standards, etc. Secondly, there was an absence of data from a sufficiently wide range of countries with diverse food supplies and in a variety of nutritional states (famines, refugees, dryland agriculture, droughts etc). 

#### 2.3.2. Cord Blood and Umbilical Cord Artery DHA Levels

Sanders and Reddy [54] reported the PUFA proportions in umbilical cord plasma and umbilical cord artery, collected at parturition, from Hindu vegetarians and from omnivores of European descent, all living in London. The data showed significantly lower DHA proportions in the vegetarian samples compared with those from omnivores (*p* < 0.001). Furthermore, the 22:5*n*-6 levels were significantly higher in the vegetarians in both samples (*p* < 0.001) and the 22:5*n*-6/DHA ratios were higher in the vegetarian than in the omnivore samples, as shown in Table 4. The infants born to the Hindu women were lighter, shorter and had smaller head circumferences than the infants born to the omnivorous mothers (*p* < 0.01). In the umbilical cord plasma, the ratio of 22:5*n*-6/DHA in the vegetarians was 0.59/1 compared with 0.27/1 in the omnivores; while in the artery the ratios were 1.02/1 in the vegetarian sample compared with 0.55/1 in the omnivores. In this example, the raised 22:5*n*-6 together with reduced DHA levels suggests there is a significant tendency towards *n*-3 PUFA depletion in this population of vegetarian women.

There was a paucity of data in the literature on equivalent samples of cord blood from other populations who might be at risk of being deficient in *n*-3 PUFA.

A prospective observational cohort study of 1838 pregnant women in India reported that higher dietary intakes of saturated fatty acids and *n*-3 PUFA were associated with increased birthweight and reduced incidence of babies who were small-for-gestational-age [55]. The combination of low ALA and low LC *n*-3 PUFA was associated with substantially increased odds of having a small for gestational age baby. The authors concluded that Indian women who were largely vegetarians would likely benefit from increased consumption of saturated fatty acids and *n*-3 PUFA and decreased consumption of *n*-6 PUFA.

#### 2.3.3. Adipose Tissue

In animals maintained on diets with ALA, but no DHA, the adipose and muscle contained DHA in more than double the amounts present in the brain DHA [56]. The source of this adipose DHA is most likely from synthesis of DHA in the liver from ALA and from there the DHA would be transported via plasma lipoproteins to various tissues such as muscle, adipose tissue, mammary gland, brain and retina. In data from the analysis of foetal tissues from infants (from western countries) reported that at term (40 weeks), it was found that 50% of the total body DHA was in adipose tissue, 23% in brain and 21% in skeletal muscle [57].

In vegans/vegetarians, it is relevant to ask whether the reduced levels of DHA observed in the blood were associated with reduced levels of DHA in adipose tissue. It is assumed that there would be sufficient ALA in the diets of such individuals to provide for some liver DHA synthesis followed by transport of the DHA via lipoproteins to adipose tissue. 

In a review of fatty acids in blood, plasma, platelets and adipose tissue, Hodgson et al. [58] found that adipose DHA levels varied from 0.0–0.28% of total fatty acids (n = 19 studies). It is possible that the studies with a zero value for DHA could be related to technical difficulties measuring fatty acids present in low concentrations. Adipose tissue samples from Seventh Day Adventists study (n = 840) in USA were reported to contain DHA from 0.10 to 0.21% total fatty acids (Table 5). They found that adipose tissue DHA levels from vegans and lacto-ovo vegetarians were 0.12% of total fatty acids, which was significantly lower than that in omnivores (0.18%) [59].

It begs the question to ask whether adipose DHA is available to the circulation for delivery to other tissues. A partial answer to this question comes from a study which reported that ALA, AA, EPA and DHA were significantly released from human adipose into the plasma FFA pool after an overnight fast [60], suggesting that adipose DHA would be available for uptake by other tissues such as mammary tissue and brain.

#### 2.3.4. Milk DHA

If the adipose in vegans/vegetarians contains DHA, then this suggests that the milk of such individuals will also contain DHA, and that this will contribute to supplying DHA to the infant brain. 

A review study of milk from 41 different countries (78 separate studies) reported that milk contained variable amounts of DHA, ranging from 0.1% (Sudan) to 0.8% (Malaysia) of total fatty acids [61]. The DHA levels were significantly higher in women who had access to marine foods compared with those who did not (0.35% vs. 0.25%, respectively, *p* < 0.05). Women from Asian countries showed the highest DHA levels amongst all the samples (Table 6). Sanders [62] reported significantly lower milk DHA values from vegan women (0.14%) compared with vegetarian (0.30%) and omnivorous (0.37%) women who were living in London. In contrast, data from USA did not reveal significantly different milk DHA results between vegans, vegetarians, and omnivores, with values ranging from 0.14 to 0.18% total milk fatty acids [63]. Two studies in Sudan reported very low DHA and low ALA values in milk from women in both North and women displaced from Southern Sudan [64,65]. These data are significant for the Sudanese infants since breast milk is the sole source of nutrition for infants for the first few months of life and weaning foods are unlikely to contain *n*-3 PUFA [66]. The traditional Sudanese diet is reported to be rich in starchy foods such as sorghum bread and porridge, rice, potatoes, and some meat [64].

#### 2.3.5. Are the Low Levels of DHA in Some Breast Milk Samples Sufficient to Support the Accretion of DHA in the Infant Brain?

Kuipers et al. [57] estimated the accretion rates of different PUFA into foetal tissues. The DHA accretion into the whole body was 41.7 mg/day in foetus of 35–40 weeks gestational age (for Western foetus’s). The question is what level of milk DHA would support this accretion rate? Using values of infant milk intake of 800 mL/day, milk fat content of 3% and DHA% of milk fat of 0.2%, it can be seen that DHA intake would be 48 mg/day. If the milk contained only 0.06% DHA (as reported above in North Sudanese women), the maximum DHA intake would equal a whole-body accretion rate of 14.4 mg/day, which is significantly below the required whole body DHA accretion rates estimated by Kuipers et al. [57].

We know from brain PM data that the DHA in the human brain increases continuously from 27 weeks gestation to about 20 years of age [67]. For this brain DHA accretion, the DHA must originate from food or endogenous synthesis from ALA or a combination of both. The steepest rise in brain DHA accumulation reported by Martinez and Moughan [68] was from 40 weeks of gestation until many months later. Clearly DHA (and/or ALA) must be supplied in the diet of the infant in this critical period of brain DHA accumulation.

#### 2.3.6. Do Post-Mortem Brain Samples Show Evidence of DHA Deficiency?

There have been relatively few studies of brain DHA and 22:5*n*-6 in humans. Most of those reported are from USA, Europe with almost no other data from the remainder of the world. Obtaining brain tissue has obvious difficulties and there are always concerns about how long after death the tissue was collected and the nature of the storage [69]. These issues have been reviewed in an elegant paper by McNamara and Jandacek [70].

##### Biochemical Indication of *n*-3 PUFA Deficiency in Infant Human Brain

Early studies by Martinez and colleagues [68,71,72], described the accretion of fatty acids into the developing human brain and retina, though the diets of these infants were not described. The proportions of arachidonic acid and adrenic acid (22:4*n*-6) in the phosphatidylethanolamine (PE) fraction and ethanolamine plasmalogens of developing human forebrains of infants aged 24–42 weeks significantly exceeded that of DHA, however with increasing age the proportion of DHA rose to levels where DHA exceeded arachidonic acid in the PE fraction. These data also showed that the proportions of 22:5*n*-6 declined from 4.31% (ages 26–42 w gestational age), to 3.55% (0–6 m) and to 2.64% (6 m to 8 y) [68,72]. 

##### Biochemical Indication of *n*-3 PUFA Deficiency in Young Children and Adolescents 

There is a paucity of data on post-mortem neural DHA levels, especially in young children and adolescents from anywhere in the world, apart from one publication from the US by Carver et al. [67] in which they reported data from 2 to 90 years (n = 88), with 19 data points between 2 and 20 years. This showed that DHA increased from <10% for the youngest subjects up to about 12–15% by the age of 20 years and then the DHA stabilized at this level up to age 90 years. This publication reported 22:5*n*-6, which was highest at early ages (2–4%) and decreased by age 20 years to approximately 2% (range 1.5–3.0%). The ratio of 22:5*n*-6/DHA was approximately 0.32 at youngest age and this declined and then stabilized to approximately 0.18 from age 20 years onwards to 90 years. In comparison, in animal studies of ALA-deficiency, brain 22:5*n*-6/DHA ranged from 1.7/1 to 9.3/1 (Figure 3).

##### Biochemical Indication of *n*-3 PUFA Deficiency in Adult Human Brains

There are limited data on the levels of DHA and 22:5*n*-6 in the adult human brain. In 1968, Svennerholm [73] reported on the fatty acid composition from 11 human brains from subjects ranging in age from 12 weeks gestational age to 82 years. The grey matter ethanolamine phosphoglyceride fraction showed that DHA increased with age from 10.8% to 33.9% and 22:5n6 increased from 2.8 to 5.0% up to 7months of age and then declined to less than 2% by 16years of age. The ratio of 22:5*n*-6/DHA from 12weeks gestational age to 7 months of age was from 0.26/1 to 0.30/1, then the value declined to less than 0.12/1 after 4 years of age. 

More recently, research studies have been conducted comparing DHA proportions of total fatty acids in various regions of the brain from control subjects with patients in whom there had been a diagnosis of either schizophrenia, Alzheimer’s disease, bipolar disorder, major depressive disorder, or Parkinson’s disease and from people who had suicided. We found 20 of these publications which reported data on the proportions of PUFA in adult human brains as shown in Table A1. Most of the samples were from USA and Canada, with others from Japan, UK, Sweden and Spain. Of the 20 publications, only 11 reported 22:5*n*-6 levels. In only 2 studies were there significant differences in DHA proportions between patients and control subjects [74,75]. Furthermore, in the studies which reported 22:5*n*-6, there were no significant differences between the 22:5*n*-6/DHA in patients compared with control subjects (Table A1). 

In Figure 3, the neural and retinal 22:5*n*-6/DHA ratios are shown for the human brain samples from infants and adults as well as from the animal studies of *n*-3 deficiency which reported changes in visual function and memory and learning. This figure reveals that none of 22:5*n*-6/DHA ratios in the human studies were in the range that have been reported for *n*-3 fatty acid deficiency in animal studies. These data showed that no evidence of *n*-3 fatty acid deficiency has been detected in adult human brain samples.

There are several limitations in the data from the published post-mortem studies, apart from those related to degradation of PUFA in the interval between death and freezing of the brain samples [70]. The additional concerns are that the data were predominantly expressed as % of total fatty acids. Reporting data this way accounts for changes in proportions relative to the total fatty acids measured, but it does not show changes in concentrations of indidual fatty acids. To account for this, studies need to include internal standards so data can be expressed as a concentration (mg per gram tissue). Another limitation is that few reports have separated the main lipid classes and determined the fatty acid concentrations in these classes, which is important for the brain since DHA is concentrated in PE and PS fractions.

##### Variability in the PUFA Proportions in Brain Regions

Systematic studies in the animal literature have reported that the fatty acid composition including DHA and 22:5*n*-6 varies with region of the brain [76,77]. Carrie et al. [76] examined the fatty acid composition of 9 regions of adult mouse brain from control and ALA deficient animals. The DHA proportions varied significantly across brain regions from 2.2% of total fatty acids in pituitary to 22.1% in frontal cortex. 22:5*n*-6 proportions also varied significantly in the deficient animals, from 2.5% to 6.7%. The pituitary lost the most DHA (control 6.6% to 2.2% in deficient) while the hippocampus showed the greatest increase in 22:5*n*-6 (control 0.1% to 4.8% in deficient). This publication illustrated the important point that there were substantial regional differences in the 22:5*n*-6/DHA ratios in the brain which is likely relevant when comparing between human PM studies. 

Hopiavuori et al. [77] reported the molecular species of phosphatidylcholine (PC), PE and PS from seven regions in the brain of mice of three different ages (2, 10 & 26 months) as determined by triple quadrupole mass spectrometry. The brain regions and lipid classes showed distinctly different molecular species profiles. This paper illustrated the importance of looking in detail at molecular species within lipid classes as the DHA molecular species varied considerably between lipid classes, with the highest concentrations being found in PE and PS fractions. Interestingly, age-related losses of DHA-containing molecular species were reported in 6 out of 7 regions and consistently in the PS fraction (especially PS 40:6), with the DHA losses being accompanied by increases in saturated and monounsaturated species. 

A landmark study in primates reported the brain PUFA composition in 26 different regions of the neonatal (4-week-old) baboon brain [31], showing that the DHA proportions ranged from 4.5% in the optic nerve to 15.8% in the globus pallidus (grey matter) region. The 22:5*n*-6 levels were reported in three regions, and these were less than 2% of total fatty acids. These data suggest that any studies on human post-mortem samples may be hard to compare between studies unless precisely the same regions were sampled for tissue analysis.

A recent paper on Huntington’s Disease (HD) separated the lipids from five major regions of the brain into lipid classes and then determined the molecular species of lipids within each class [78]. The molecular species of the glycerophospholipids were separated by nano electrospray ionisation mass spectrometry which yielded major lipid species such as PC 18:1–22:6, PE 18:0–22:6, etc. In the HD samples, there were substantial decreases in the concentrations of PC in three areas of the brain (white cortex, grey cortex, putamen), and also substantial decreases in PE (putamen) and in ether PE and PS (white cortex). These decreases in the specific lipid classes in the HD samples ranged from 18 to 37%. In the HD putamen, the 22:6-containing molecular species were significantly lower in all PL classes (42% for ester linked and 13% for ether PL), especially in PE and PS fractions where losses of between 26% and 50% were recorded for certain DHA molecular species. Molecular species containing 22:5 were reported, however there was no differentiation between 22:5*n*-6 and 22:5*n*-3 isomers. The ratio of total 22:5/DHA did not exceed 0.2 in any class where significant losses of DHA were observed. The authors speculated that the decreases in DHA could be related to increased formation of inflammatory mediators from DHA or a change in the transport of DHA into the brain via the lysophosphatidylcholine carrier. 

## 3. Discussion

The aim of this paper was to identify evidence for a dietary deficiency of DHA in human tissues, and in human brain samples collected at post-mortem. We used elevations in the ratio of 22:5*n*-6/DHA (*n*-3 PUFA deficiency score) as evidence there was a dietary-induced DHA deficiency in tissues such as plasma/blood, adipose, milk, artery tissue, and brain, with22:5*n*-6/DHA ratios > 1/1 associated with >50% depletion of neural DHA [19]. In vegan and some vegetarian populations, reduced concentrations of DHA in plasma, blood and tissues, including adipose tissue and milk, were found. Vegetarian diets were typically high in LA and low in ALA and lacking LC *n*-3 PUFA. In most of these studies, there was no evidence of significantly increased levels of the biochemical marker of *n*-3 PUFA deficiency, namely 22:5*n*-6. While vegans were at risk of low DHA levels in blood and tissues, they had DHA in their adipose tissue and milk within the range for other diet groups.

We found three examples which pointed to *n*-3 PUFA deficiency, all related to the new-born infants. Firstly, raised 22:5*n*-6/DHA deficiency indicator status of cord arterial tissue from Hindu vegetarian women living in London (UK). Secondly, data from the UK and Australia showed raised 22:5*n*-6/DHA deficiency indicator status of new-born infants when the infants were fed with low *n*-3 PUFA formulas. Thirdly, milk samples from women from northern Sudan which showed very low milk DHA levels.

In a contrast to the present findings, there has been much discussion in the literature about the need for *n*-3 PUFA supplements based on low levels of DHA in blood and associations of these levels with various neurological conditions, as outlined recently by von Schacky [5]. This has contributed to enormous growth in the sale of *n*-3 PUFA supplements, throughout the world, currently in excess of USD1.54 bill (2020 sales data) which has been predicted to grow at a compound annual growth rate of 7.8% until 2028 [79].

### 3.1. Plasma, Red Blood Cells, Adipose, Milk and Artery Data

There were many publications in the literature, mostly from westernized countries, which consistently highlighted that vegetarian and vegan subjects had low blood and tissue DHA levels compared with omnivores. Some of these analyses for vegan subjects reported moderately raised 22:5*n*-6 values in plasma and red blood cells compared with omnivores, associated with significantly lower DHA values. The 22:5*n*-6/DHA (deficiency score) did not exceed a value of 0.4. To put this in context, in animal studies of *n*-3 PUFA deficiency associated with changes in visual or neural function, 22:5*n*-6/DHA values in brain and retina ranged from 1.2/1 to 9.3/1 (Figure 3). The samples showing the greatest elevation in the deficiency score were from Hindu vegetarian subjects living in London, data which was published in the 1970s. This revealed that cord blood and cord arterial tissue had 22:5*n*-6/DHA values almost double those from infants born to omnivorous mothers. The highest deficiency ratio (1.02/1) was in the cord artery of vegetarian subjects compared with 0.6/1 in the cord artery of the omnivorous subjects.

The adipose tissue and milk samples from vegetarians and vegans typically reported DHA values significantly lower than those from omnivorous subjects, however, the 22:5*n*-6 values were rarely reported. The lowest milk DHA values were recorded in mothers from North Sudan (0.6% fatty acids) and may not have been sufficiently high to sustain typical DHA accretion rates into the infant brain based on data from Kuipers et al. [57].

### 3.2. Brain Data

There were fewer than 30 publications which reported the levels of fatty acids in post-mortem samples of human brain. The data can be grouped in several ways, [a] descriptive data on the fatty acid composition of brain samples taken from foetal brains up to adult brains (90 years), where no dietary data was reported, [b] data on foetal and neonatal brains where diets of either mothers or the neonatal infants (infant formulas and breast milk) were available, and [c] data from adults who had known neurological conditions compared with samples from controls (age and gender-matched), where no dietary data was reported. 

[a] Descriptive data (healthy individuals)—dietary details unknown: Four publications provided data on the developmental changes in 22:5*n*-6 and DHA levels in post-mortem brain samples from 12-week-old foetus to 90-year-old individuals [40,67,68,73]. These data were gathered in Tanzania, USA, Spain and Sweden, respectively. There was a consistent trend in these publications for DHA levels to increase with age from foetal samples to about 20–30 years of age and then for the DHA levels to plateau. In contrast, for 22:5*n*-6 which was present across the life span, the levels declined with age and plateaued by about 20–30 years. In the earliest samples (foetal) the 22:5*n*-6 was between 24% and 47% of the DHA level, whilst in the adult samples 22:5*n*-6 levels declined to approximately 14% of the DHA level. 

[b] Descriptive data—dietary details known (new-born): Two publications, one from Scotland and one from Australia, were found for neonatal infants and these reported significantly raised ratios of 22:5*n*-6/DHA in brain tissue of neonatal infants maintained on infant formula milks with high levels of LA and low ALA (in one formula, the LA/ALA was 36/1) compared with infants maintained on breast milk [27,29]. In the brain samples of the formula-fed infants, the mean DHA value was only 60% of that of the breast-fed infants in the most sensitive PL fraction, and the 22:5*n*-6/DHA was 0.72/1 in these infants [27]. The longer the infants were on the low ALA formulas, the higher the 22:5*n*-6 levels and lower the DHA levels.

[c] Descriptive data—neurological conditions—dietary details unknown: Analysis of post-mortem brain lipid samples from adults with various neurological disorders, reported in 20 studies from the developed world, revealed no evidence of elevated levels of 22:5*n*-6, even if the samples showed reduced DHA levels compared with control subjects. For the samples from subjects with neurological disorders, the 22:5*n*-6 levels ranged between 6% and 16% of the DHA levels of the control subjects, whether or not there were reduced DHA levels.

The brain data revealed that 22:5*n*-6 is a normal constituent of human brain, and that its levels declined with age, particularly from before birth and in the first 20 years of life. The brain data also revealed that dietary changes in postnatal infants with low ALA formulas, rich in LA, resulted in significantly reduced DHA levels and raised 22:5*n*-6 levels, and that the 22:5*n*-6 increased the longer the infants were maintained on the low ALA formulas. This suggests that human brain fatty acid levels were responding to low ALA diets in a similar manner to animals on diets low or deficient in *n*-3 PUFA. Thirdly, there was no evidence of raised 22:5*n*-6/DHA in brain samples taken from adult subjects with various neurological conditions.

#### 3.2.1. What Are the Implications and Limitations of the Brain Data?

Most samples analysed were from subjects/patients from western countries. What are the implications of this? Most western countries have access to foods containing *n*-3 PUFA, whether from land-based sources such as seed oils, plant leaves, nuts, meat and eggs, as well as from seafood. In addition, the use of *n*-3 PUFA supplements is widespread in many westernised countries and in developing countries fish are commonly consumed [79]. 

However, there are likely to many areas of the world which have severely limited intakes of *n*-3 PUFA, owing to low or minimal intakes of fish (marine), limited intakes of meat/offal, limited intakes of eggs from chickens with access to green feed, use of vegetables oils with very low ALA levels (sunflower, peanut/groundnut oil etc) and limited intakes of green leafy vegetables. 

#### 3.2.2. What Are the Most Likely Places and Conditions to Find Dietary Insufficiency of *n*-3 Fatty Acids?

Two publications were identified which offered indications of where there might be an *n*-3 insufficiency. One publication was from Indian schoolchildren [53], a country with a high proportion of vegetarian subjects and with substantial poverty in rural areas [80]. The other publication reported the breast milk fatty acid levels from the Horn of Africa including women from northern Sudan [64]. The diet of these women was LA-rich fermented sorghum bread, LA-rich ALA-poor vegetable oils (cottonseed and groundnut oils) and flat-leafed green vegetables [65]. 

In addition to these regions of the world, poverty is still widespread throughout the world and the lack of adequate food intakes has been exacerbated by continuing conflict in many regions. Forced migrations from refugees fleeing from conflict, persecution and climate change (flooding, droughts, fires) are other groups of individuals whose food supply is likely to be highly compromised. Estimates from UNHCR [81] suggest that there are more than 100 million forcibly displaced people living in poverty, including those living in refugee camps, suffering from effects of excessive climatic events (droughts, fires, floods), and more than 10 million living under conditions of continuing conflict. We suggest all these people may have limited access to foods containing *n*-3 PUFA and that they would be vulnerable to *n*-3 PUFA dietary insufficiencies.

#### 3.2.3. What Are the Conditions and Situations Where Brain DHA Levels Are Vulnerable to Depletion?

Based on the published data on the accrual of DHA into the human brain, the most vulnerable periods are during foetal and postnatal development up to about 2 years of age [40,71]. That is not to exclude later age groups, such as weanling infants, young children, and adolescents, as DHA levels in the brain continue to rise until about 20 years of age. In addition, there is a need throughout life for continual replacement of the DHA due to turnover [82]. The conditions and situations where the brain DHA levels are vulnerable to depletion are depicted in Table 7. To put some numerical values on the intakes of PUFA referred to in Table 7, the generally recommended intakes for ALA are 1 g/day [46] and for LCn-3 PUFA 250 mg/day [47]. High intakes of LA relative to ALA would refer to situations where the main vegetable oils used would have LA/ALA of >50:1, such as found in cottonseed oil, ground nut oil, and sunflower oil [51].

#### 3.2.4. Unknowns 

We know that animals fed diets lacking ALA showed significantly reduced capacity to learn and have reduced visual, olfactory, and auditory function compared with those fed ALA or ALA + DHA. These animals had significantly reduced DHA in the brain and retina. We do not know whether these physiological changes are in a linear relationship with DHA content of neural membranes. There is almost no data which has systematically addressed the relationship between graded levels of ALA (+DHA) in the diet leading to graded levels of neural DHA and at the same time measuring neural, olfactory, auditory and retinal function. 

However, one study in guinea pigs determined the retinal response to light against graded levels of retinal DHA induced by dietary *n*-3 PUFA deprivation [83]. In this study, the sensitivity of the retinal response was significantly affected by dietary-induced deficiency of retinal DHA. Animals with a 25% DHA loss or greater (22:5*n*-6/DHA of 0.46/1 or higher) showed significant losses in retinal sensitivity compared with the control retinal DHA values. 

This is an important question to answer in future studies as it should be a relevant physiological reference point for understanding the impact of DHA losses in the neural system in humans.

## 4. Conclusions

DHA is of fundamental importance for optimal neural and visual function. This review systematically searched for evidence of deficiency of DHA in human populations. We concluded that rather than there being evidence of the absence of dietary *n*-3 PUFA deficiency in human brain samples, there is an absence of evidence because of a lack of data from populations deemed to be most likely at risk of *n*-3 PUFA deficiency. Currently available data were skewed towards westernised countries where dietary *n*-3 PUFA deficiency was less likely to exist.

In vegan and some vegetarian populations, reduced concentrations of DHA in plasma, blood and tissues, including adipose tissue and milk, were found. Vegetarian diets were typically high in LA and low in ALA and lacking LC *n*-3 PUFA. In most of these studies, there was no evidence of significantly increased levels of the biochemical marker of *n*-3 PUFA deficiency, namely 22:5*n*-6. While vegans were at risk of low DHA levels in blood and tissues, they had DHA in their adipose tissue and milk.

We found three examples which pointed to *n*-3 PUFA deficiency, all related to new-born infants. Firstly, raised deficiency indicator status of cord arterial tissue from Hindu vegetarian women living in London (UK). Secondly, data from the UK and Australia showed raised deficiency indicator status of new-born infants when the infants were fed with low *n*-3 PUFA formulas. Thirdly, milk samples from women from northern Sudan showed very low milk DHA levels.

There is a huge gap in the literature on data on the DHA and 22:5*n*-6 status in brain tissue from different regions of the world which are considered at risk of low *n*-3 PUFA intakes. We suggest that data should be gathered from populations regarded at risk of dietary *n*-3 PUFA deficiency (dryland agriculture, regions of famine, floods, war situations and refugees in camps or on the move) and from individuals in vulnerable life stages, namely weanling infants, children, and adolescents.

## Figures and Tables

**Figure 1 nutrients-15-00161-f001:**
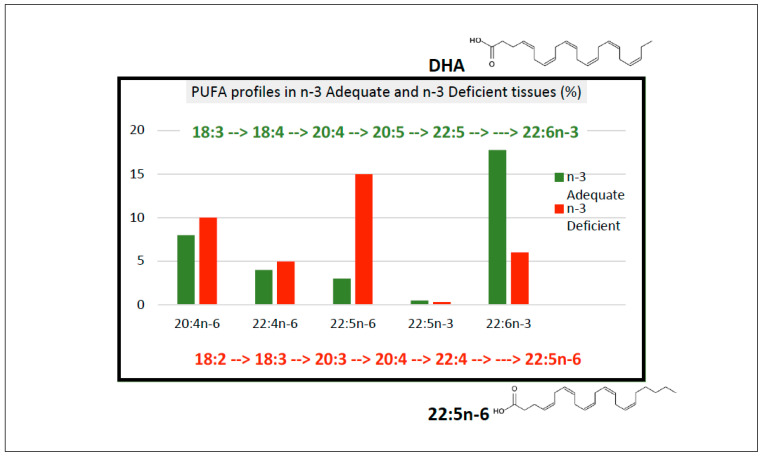
Metabolism of *n*-3 and *n*-6 PUFA, structures of DHA and 22:5*n*-6 and PUFA profiles in *n*-3 Adequate and *n*-3 Deficient tissues.

**Figure 2 nutrients-15-00161-f002:**
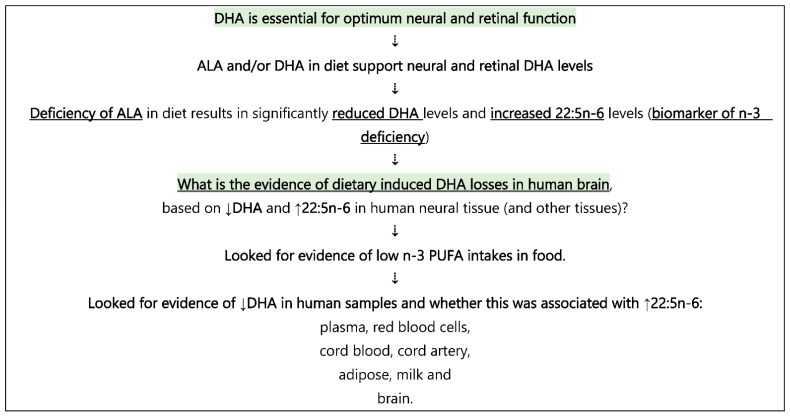
Strategy and logic of the data presented in this paper.

**Figure 3 nutrients-15-00161-f003:**
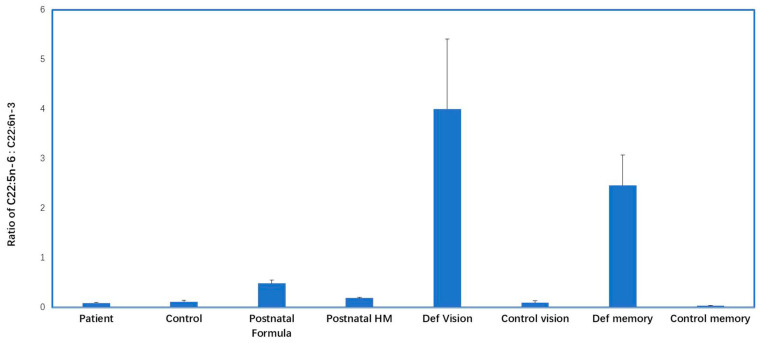
The ratio of 22:5*n*-6 to DHA (22:6*n*-3) in infant and adult human brains compared with the brain and retina of ALA-deficient animals and ALA-adequate animals. The data are shown as mean ± SD for the human and animal samples. Patient: post-mortem data for adult brain (n = 18) with schizophrenia, depression, Alzheimer’s disease, Parkinson’s disease, suicide, see Table A1 for reference details. Control: post-mortem data for adult brain from healthy control subjects, see Table A1 for details (n = 12). Postnatal formula: post-mortem data from infants aged up to 12 months fed infant formulas with high LA:ALA ratios (n = 11), (Farquharson et al. [27]). Postnatal HM: post-mortem data from infants aged up to 12 months fed human milk (n = 5), (Farquharson et al. [27]). Deficient vision: retina from animals maintained on *n*-3 deficient diets where visual function was assessed (n = 5 studies), (Bush et al. [30]; Diau et al. [31]; Niu et al. [32]; Weisinger et al. [33]; Weisinger et al. [34]). Control vision: retina from animals maintained on *n*-3 adequate diets (n = 5 studies), [30,31,32,33,34]. Deficient memory: brains from animals maintained on *n*-3 deficient diets where learning and memory was assessed (n = 5 studies), (Federova et al. [35]; Frances et al. [36]; Lamptey and Walker [37]; Lim et al. [38]: Moriguchi et al. [39]). Control memory: from animals maintained on *n*-3 adequate diets where learning and memory was assessed (n = 5 studies), [35,36,37,38,39].

**Table 1 nutrients-15-00161-t001:** Conditions associated with reduced neural DHA levels.

	Condition	Description
1	Dietary deficiency of ALA	Dietary insufficiency of ALA which would be expected to result in decreased DHA synthesis in the liver and increased 22:5*n*-6 synthesis from LA, and hence less DHA and more 22:5*n*-6 transported to brain via plasma lipoproteins.
2	Brain DHA uptake receptors	Polymorphisms in brain fatty acid transport proteins such as fatty acid binding proteins (FABP), fatty acid transport proteins (FATP), fatty acid translocase FAT/CD36, DHA uptake receptors Mfsd2a, and APOE4, and hence less uptake of DHA into the brain from the circulation [7,8].
3	Brain DHA metabolism	Acyl CoA synthetase 6 controls neuronal DHA levels independent of diet by ligating CoA to free DHA, and hence polymorphisms could reduce brain DHA levels [9]
4	Brain PUFA peroxidation	Increased peroxidation of PUFA in brain, where it would be expected to result in increased indices of lipid peroxidation, resulting peroxidation/loss of DHA in the brain [10].
5	Liver ALA metabolism	Polymorphisms in liver FADS1/2 desaturases and elongases [11,12,13] resulting in reduced DHA synthesis from ALA, and hence less DHA transported to brain via plasma lipoproteins.
6	Liver function	Liver damage, such as fatty liver induced by diet, alcohol, or liver diseases, which might impact on the capacity of the liver to synthesise DHA from ALA, and hence less DHA transported to brain via plasma lipoproteins [14,15].

**Table 3 nutrients-15-00161-t003:** Plasma and red blood cell polyunsaturated fatty acids in vegans and omnivores (% total fatty acids).

Fatty Acid %	Vegan Plasma PC ^1^	Omni Plasma PC ^1^	Vegan RBC ^2^	Omni RBC ^2^	Vegan RBC PS ^3^	Omni RBC PS ^3^	Vegan Plasma Total ^4^	Omni Plasma Total ^4^	School Age (M) Plasma PL ^5^	School Age (F) Plasma PL ^5^
**18:2*n*-6**	33.3	26.0	14.2	11.6	3.3	2.2	33.1	27.1	27.2	27.5
**20:4*n*-6**	10.6	9.1	14.3	13.7	28.5	23.1	6.6	6.7	10.0	9.3
**22:4*n*-6**	0.6	0.3	3.0	2.0	3.6	2.1	0.2	0.2	2.7	2.9
**22:5*n*-6**	0.4	0.2	0.5	0.2	1.2	0.6	0.2	0.3	0.6	0.6
**18:3*n*-3**	nr	nr	0.1	0.1	0.0	0.0	0.7	0.5	nr	nr
**20:5*n*-3**	0.2	1.3	0.5	1.4	0.4	0.9	0.5	1.0	nr	nr
**22:5*n*-3**	0.8	1.1	2.1	2.5	3.4	4.0	0.5	0.6	0.2	0.2
**22:6*n*-3**	1.4	4.0	3.3	6.7	7.8	14.1	0.9	2.2	1.0	1.0
**22:5*n*-6/DHA**	0.29	0.05	0.15	0.03	0.15	0.04	0.26	0.12	0.6	0.6

^1^ Sanders et al. [52] plasma PC from UK; ^2^ Agren et al. [44] RBC total FA from Finland; ^3^ Agren et al. [44] RBC-PS FA; ^4^ Pinto et al. [43] plasma total FA from UK; ^5^ Parasannanavar et al. [53] plasma PL FA from Indian school children. PC = phosphatidylcholine. PS = phosphatidylserine. RBC = red blood cell. Omni = omnivore. PL = phospholipid.

**Table 4 nutrients-15-00161-t004:** Cord plasma and cord artery DHA and 22:5*n*-6 levels from vegans and omnivores (% of total fatty acids).

Fatty Acid	Cord Plasma Vegetarian	Cord Plasma Omnivore	Cord Artery Vegetarian	Cord Artery Omnivore
22:5*n*-6	2.34 ± 0.16 ^a,1^	1.58 ± 0.13 ^b^	4.15 ± 0.16 ^a^	3.19 ± 0.15 ^b^
22:6*n*-3 (DHA)	4.00 ± 0.36 ^a^	5.84 ± 0.31 ^b^	4.05 ± 0.17 ^a^	5.75 ± 0.19 ^b^
22:5*n*-6/DHA	0.59	0.27	1.02	0.55

^1^ Values with different superscript letters within a tissue are significantly different (*p* < 0.01) Data from Sanders and Reddy [54].

**Table 5 nutrients-15-00161-t005:** Adipose tissue fatty acids in vegans, lacto-ovo vegetarians and non-vegetarians (% total fatty acids) ^1^.

Population	LA	AA	ALA	EPA	DPA *n*-3	DHA	LA/ALA
Vegan n ≥ 67	23.3 ^a,2^	0.39 ^a^	1.6 ^a^	0.02	0.21 ^a^	0.12 ^a^	14.6 ^a^
LO-veg n ≥ 218	21.7 ^a^	0.44 ^a^	1.2 ^a^	0.02	0.20 ^a^	0.12 ^a^	18.1 ^b^
Non-veg n ≥ 374	19.1 ^b^	0.51 ^b^	1.0 ^b^	0.03	0.24 ^b^	0.18 ^b^	19.1 ^b^

^1^ Data from Miles et al. [59]; ^2^ LO-veg = lacto-ovo vegetarian. Non-veg = non-vegetarian. ^2^ Values with different superscript letters within a fatty acid are significantly different (*p* < 0.01).

**Table 6 nutrients-15-00161-t006:** Human milk fatty acids (% total fatty acids).

Population	LA	AA	ALA	EPA	DHA	LA/ALA	Total *n*-3
Vegans ^a^	23.8	0.32	1.36	nr	0.14	17.5	1.50
Vegetarians ^a^	19.5	0.38	1.25	nr	0.30	15.6	1.55
Omnivores ^a^	10.9	0.35	0.49	nr	0.37	22.2	0.86
South Sudan ^b^	14.7	0.60	0.28	0.04	0.10	52.5	0.48
North Sudan ^c^	12.8	0.48	0.20	0.05	0.06	64.0	0.36

^a^ Sanders [62]; ^b^ Nyuar et al. [65]; ^c^ Nyuar et al. [66]. nr = not reported.

**Table 7 nutrients-15-00161-t007:** Conditions and situations where the brain DHA levels are vulnerable to depletion.

Vulnerable Lifespan Periods	Vulnerable Diets	Vulnerable Nutrient Situations	Vulnerable Populations
Multiple pregnancies, Postanal infants, Weanling Infants, Children, Adolescents	High LA vegetable oils, No green vegetables, No eggs, No fish/shellfish, No fresh milk and meat	High LA/ALA, Low ALA, No LC *n*-3 PUFA	Dryland agriculture, Deserts, Famines, floods, fires, Refugees

## Data Availability

Not applicable.

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
