# Peer review of "What Is the Evidence for Dietary-Induced DHA Deficiency in Human Brains?"

_nutrients, 2022, doi:10.3390/nu15010161_

Round 1

Reviewer 1 Report

1. The meaning of current research is high; However, those sentences are difficult to understand.

2. The content has too many aspects and it is difficult to follow. It is better to give a logic to follow, otherwise, it easily gets lost.

3. The content has no connection with each other, it is better to give a chart to illustrate what kind of aspects the author would like to describe the DHA and how is the interconnection between them. 

Reviewer 2 Report

This is a very comprehensive review on the worldwide epidemiology of DHA deficiency covering the most important features of this topic.

The Results section reads somewhat uninspiring due to numerous numbers and references but this is actually the nature of such large reviews. This could probably be cured by displaying some data in Tabular form and by adding a few more  figures.

SPECIFIC COMMENT

1.     It is suggested to include in the text some cartoons displaying the pathway of DHA biosynthesis from ALA with the corresponding structure of the compounds.

2.     The firs 22 lines of ¶ 3. (Discussion) might fit better in the Introduction. It would help the reader to understand the following chapters if the authors would list all the components (enzymes) involved in DHA metabolism in their very first part of the manuscript preferentially by mentioning the particular pathways . Part of them might be shown as cartoons.

3.     For the sake of practicability the authors should add a paragraph on the daily recommended doses of DHA, ALA and possibly EPA.

Round 2

Reviewer 1 Report

No

Reviewer 2 Report

The manuscript has improved significantly ant there are no more suggestions from my side.